# Transcending the slow bimolecular recombination in lead-halide perovskites for electroluminescence

Guichuan Xing[1], Bo Wu[2], Xiangyang Wu[3], Mingjie Li[2], Bin Du[4], Qi Wei[4], Jia Guo[4], Edwin K.L. Yeow[3], Tze Chien Sum[2] & Wei Huang[4,5]

The slow bimolecular recombination that drives three-dimensional lead-halide perovskites' outstanding photovoltaic performance is conversely a fundamental limitation for electroluminescence. Under electroluminescence working conditions with typical charge densities lower than $10^{15}\,\mathrm{cm}^{-3}$, defect-states trapping in three-dimensional perovskites competes effectively with the bimolecular radiative recombination. Herein, we overcome this limitation using van-der-Waals-coupled Ruddlesden-Popper perovskite multi-quantum-wells. Injected charge carriers are rapidly localized from adjacent thin few layer ($n \leq 4$) multi-quantum-wells to the thick ($n \geq 5$) multi-quantum-wells with extremely high efficiency (over 85%) through quantum coupling. Light emission originates from excitonic recombination in the thick multi-quantum-wells at much higher decay rate and efficiency than bimolecular recombination in three-dimensional perovskites. These multi-quantum-wells retain the simple solution processability and high charge carrier mobility of two-dimensional lead-halide perovskites. Importantly, these Ruddlesden-Popper perovskites offer new functionalities unavailable in single phase constituents, permitting the transcendence of the slow bimolecular recombination bottleneck in lead-halide perovskites for efficient electroluminescence.

[1] Institute of Applied Physics and Materials Engineering, University of Macau, Macao SAR 999078, China. [2] Division of Physics and Applied Physics, School of Physical and Mathematical Sciences, Nanyang Technological University, 21 Nanyang Link, Singapore 637371, Singapore. [3] Division of Chemistry and Biological Chemistry, School of Physical and Mathematical Sciences, Nanyang Technological University, 21 Nanyang Link, Singapore 637371, Singapore. [4] Key Laboratory of Flexible Electronics (KLOFE) & Institute of Advanced Materials (IAM), Jiangsu National Synergetic Innovation Center for Advanced Materials (SICAM), Nanjing Tech University (Nanjing Tech), 30 South Puzhu Road, Nanjing 211816, China. [5] Key Laboratory for Organic Electronics and Information Displays & Institute of Advanced Materials (IAM), SICAM, Nanjing University of Posts & Telecommunications, 9 Wenyuan Road, Nanjing 210023, China. Correspondence and requests for materials should be addressed to T.C.S. (email: Tzechien@ntu.edu.sg) or to W.H. (email: wei-huang@njtech.edu.cn).

Since 2009, solution-processed three-dimensional (3D) lead-halide perovskites have demonstrated great potential in solar cell applications with light to electricity conversion efficiency up to 22.1% being achieved[1–3]. The primary advantages of these lead-based perovskites for photovoltaics are their large photon absorption coefficient[1], long-range balanced charge carrier diffusion lengths[4,5], low trap density[6], high charge carrier mobility[7], fast and efficient photo-generated exciton dissociation and slow electron-hole bimolecular recombination[8]. Furthermore, their widely tunable direct bandgap also makes them potential candidates for light emitting diodes (LEDs). Solution-processed LEDs are attractive for applications in low-cost, large-area lighting sources and displays. This technology has been pursued for nearly three decades with organic materials and the highest reported external quantum efficiency (EQE) is around 24% (ref. 9). Solution-processable perovskites are promising alternatives to the organic molecules for such LEDs applications. To date, respectable perovskite LED efficiencies (8.8%) with tunable emission colours in areas larger than the National Television System Committee (NTSC) standard have been realized[10–15]. However, the slow free electron-hole bimolecular radiative recombination in these 3D lead-halide perovskites (that spurs efficient photovoltaic operation) is a fundamental limitation to improving their electroluminescence (EL) efficiencies (Fig. 1a). The EQE is greatly limited due to the effective competition from the defect-states trapping (first order decay process) with the slow electron-hole bimolecular radiative recombination (second order decay process) under LED working conditions (i.e., electrical injected charge-carrier density is typically lower than $10^{15}$ cm$^{-3}$, which is comparable to the 3D perovskite trap densities).

Perovskite nanocrystals (NCs) and two-dimensional (2D) layers with first order excitonic recombination are potential solutions to this problem[16–19]. While significant progress has been made with perovskite NC LEDs, stumbling blocks include: the conflicting requirements of efficient charge injection and transporting (that is, requiring large grain size and good electronic coupling between assembled NCs) and high efficiency excitonic luminescence (that is, requiring small grain size and well-isolated NCs) in NC-assembled thin film LED, and the demanding requirement of heteroepitaxially hybridizing the NCs with a large-bandgap, interfacial trap-free conductive host in NC-doped LED. Although 2D perovskite bulk crystals[20] associated with fast excitonic emission and high charge carrier mobility are promising, daunting challenges pertaining to significant nonradiative recombination need to be effectively tackled (that is, surface trapping, intrinsic defect trapping, extremely strong exciton-phonon coupling and exciton–exciton interaction, Supplementary Note 2). Furthermore, the efficiencies of 2D perovskite LEDs are extremely low even at cryogenic temperatures[18,19].

2D organometal halide perovskites ($L_2MX_4$) comprise of monomolecular layers of [$MX_6$] octahedral sandwiched between long organic chains (L) barrier layers, where M is a divalent metal cation, X is a halide anion. The injected excitons are therefore tightly confined in the 2D ($n=1$) inorganic layers and feature large exciton binding energies and undergo significant non-radiative recombination in the 2D bulk crystals[18–23]. On the other hand, 3D perovskites ($SMX_3$) consist of infinite layers of [$MX_6$] octahedral connected at the halide, forming a 3D network with the short organic cation (S) located at the void of the network. The injected excitons are weakly confined and will rapidly and

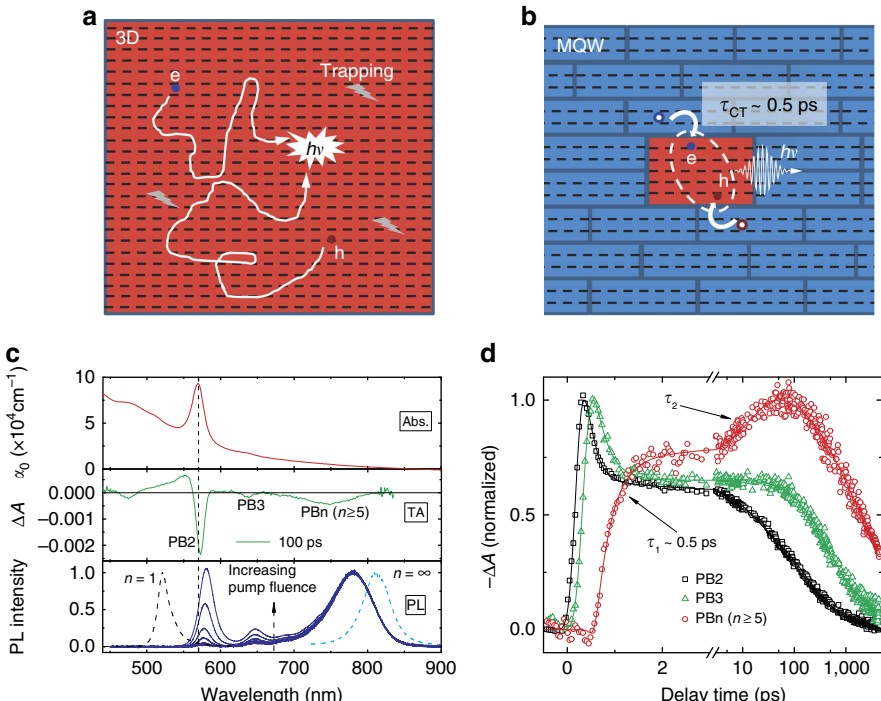

**Figure 1 | Van-der-Waals coupled perovskite multi-quantum wells (MQWs) for light emission.** (**a**) An illustration of the fundamental limitation in three-dimensional (3D) perovskite electroluminescence—defect states trapping although rare, can still be very effective in competing with the slow bimolecular recombination at low carrier excitation densities. (**b**) Illustration of charge injection, exciton localization ($\tau_{CT}$) and subsequent recombination in the perovskite MQWs from the thin QWs ($n \leq 4$, host) to the self-assembled thick ($n \geq 5$) QW (TQW) dopants. (**c**) Ultraviolet-Vis absorbance (Abs.), chirp-corrected photo-induced changes in absorption ($\Delta A$) and pump fluence dependent photoluminescence (PL) spectra (blue lines, normalized at 780 nm) for perovskite MQWs film. The TA was performed with 400 nm excitation (at 1 KHz, 100 fs, around 0.4 μJ cm$^{-2}$). The PL spectra were collected by varying pump fluence of the same laser beam from 0.002 to 7.3 μJ cm$^{-2}$. The PL spectra of 2D ($n=1$) and 3D ($n=\infty$) perovskite films (at 7.3 μJ cm$^{-2}$) are also shown here for comparison. (**d**) Normalized bleaching kinetics at PB2 (2.18 eV), PB3 (1.96 eV) and PBn (1.65 eV) as indicated in **c**.

efficiently dissociate into free electrons and holes[7,8]. In-between these 2D and 3D classes of perovskites, there is also a family of quasi-2D perovskite quantum wells (QWs) $L_2(SMX_3)_{n-1}MX_4$, also commonly known as Ruddlesden-Popper perovskites, whose connected $n$ layers of inorganic $[MX_6]$ octahedral monomolecular sheets are sandwiched by two organic barrier layers. The QWs are stacked together with atomically sharp interfaces and relatively weak van-der-Waals interactions in the bulk crystal. The excitons are quantum confined within the inorganic wells, but the quantum confinement effect decreases with increasing well width. Hence, the excitonic properties of the quasi-2D QW can be systematically tailored from 2D to 3D by increasing the inorganic layer number from $n = 1$ to $n = \infty$ (refs 21–23). Tsai et al.[24] reported that the excitonic properties of the high-quality QW crystals ($n = 3$ and $n = 4$) are very close to that of the 3D perovskite ($n = \infty$). Most recently, remarkably high-efficiency perovskite LEDs constructed with the multi-QW (MQW) assemblies (with multi-thickness QWs in one crystal film) have been demonstrated[14,15]. However, detailed understanding into the innate dynamics of the injected charge carriers in the MQWs assemblies driving the high efficiencies of these LEDs remains inadequate.

Herein, through detailed femtosecond transient optical spectroscopy, we present concrete evidence that these self-assembled lead-halide perovskite MQWs can effectively overcome the slow bimolecular recombination bottleneck for perovskite LEDs. Charge carrier localization from the thin QWs (host, $n \leq 4$) to the thicker QWs (TQWs) (dopants, $n \geq 5$) in these solution-processed van-der-Waals coupled MQWs with atomically sharp interfaces, is extremely fast (around 0.5 ps) and efficient (over 85%) due to quantum coupling[25]. The luminescence is dominated by exciton recombination in the self-assembled TQWs within the thin QW host (Fig. 1b). Its first-order exciton recombination can effectively compete with defect states trapping over a broad range of injected charge carrier densities. These mechanisms give rise to a near invariant high photoluminescence quantum yield (PLQY) (around 60%) from the TQWs where the charge carriers are primarily injected into the thin wells at carrier densities lower than $10^{16}$ cm$^{-3}$. In contrast, the PLQY for 3D perovskite decreases significantly with decreasing injected carrier densities over the same range. These perovskite thin QWs inherit the favourable properties of 2D perovskites: simple low-temperature solution processability, high charge carrier mobility and uniform morphology[14,15,26,27]. Meanwhile, the self-assembled TQWs inherit the high crystallinity and low trap density of perovskite NCs. These complementary properties in Ruddlesden-Popper phase MQW perovskites, that are typical of epitaxial vacuum-grown heterojunctions[28,29], render us new functionalities which are unavailable with traditional single-phase halide perovskites.

## Results
**Perovskite MQWs synthesis**. In this study, the Ruddlesden-Popper phase perovskite MQWs film was deposited with a precursor solution of 1-naphthylmethylamine iodide (NMAI, $C_{10}H_7CH_2NH_3I$), formamidinium iodide (FAI, $CH_5IN_2$) and $PbI_2$ with a molar ratio of 2:1:2 dissolved in N,N-dimethylformamide (DMF)[21,22]. The 2D $(NMA)_2PbI_4$ and 3D $FAPbI_3$ films were prepared with the same methods described above by adding stoichiometric precursors in DMF (40 wt% for 2D, 20 wt% for 3D; for characterization see Supplementary Figs 1–4 and Supplementary Note 1 ). With the same dimensional structure and similar sample quality of the lead-halide perovskites, different composition will result in slightly different decay parameters, but will not change the basic decay physics[30,31]. Therefore, the photophysics illustrated below is broadly applicable.

**Fast and efficient carrier localization in perovskite MQWs**. The obtained MQWs film is extremely flat and pin-hole free (Supplementary Fig. 1). The linear absorption spectrum of the film (Fig. 1c) shows a dominant exciton absorption peak located at around 568 nm (or 2.18 eV), which confirms that the primary component of the film is a lead-iodide based bilayer ($n = 2$) QW $(NMA)_2(FAPbI_3)PbI_4$ (refs 21,22). The pronounced narrow bilayer exciton absorption (FWHM around 10 nm) also suggests that the electronic coupling between the stacked QWs is relatively low. Another small but obvious absorption peak at around 633 nm (1.96 eV) indicates the presence of some low-concentration TQWs doped within. This self-doping effect can be more clearly observed from the transient absorption (TA) spectra and pump fluence-dependent PL spectra as shown in Fig. 1c (Figs 1d and 2 and Supplementary Figs 10–12).

In TA spectroscopy, the MQWs film is first excited with a femtosecond laser pulse, the photo-induced changes in absorption ($\Delta A$) spectrum are then probed with a time-delayed laser-generated white light probe pulse. $\Delta A$ is directly proportional to the change in absorption coefficient. On excitation by 3.1 eV photons, the excitons are primarily injected into the bilayer QWs. As the band gap of the QW decreases with increasing numbers ($n$) of the inorganic $[PbI_6]$ layers; the decay of these photo-injected excitons will experience a competition between the recombination within the bilayer QWs and localization from the bilayer QWs to self-assembled thicker QWs. The time-delayed exciton accumulation in thick QWs can be probed from the photo-induced bleaching (PB) in their respective exciton transitions. In addition to bilayer QW exciton peak (2.18 eV), the time-delayed TA spectrum in Fig. 1c (also in Supplementary Figs 10a and 12) distinctly shows other exciton resonances peaked at around 633 (1.96 eV), 674 (1.84 eV) and 751 nm (1.65 eV). The first two peaks agree well with previously reported values for trilayer (1.96 eV) and tetralayer (1.84 eV) $[PbI_6]$ QWs[21,22]. The third broad peak (1.65 eV) should be from the overlapped exciton resonances in QWs with inorganic layer number $n \geq 5$ (TQWs). The time evolution of the TA signals at these different exciton resonance peaks directly probes the carrier localization dynamics from thin QW to the self-assembled TQWs. Figure 1d presents the dynamic evolution of the bilayer, trilayer and TQWs exciton resonances in the self-assembled MQWs. We found that the prominent PB at bilayer exciton resonance appears instantaneously, which confirms that the excitons are directly created in the bilayer QWs. Following the fast buildup, the PB at bilayer exciton resonance shows a dominant ultrafast decay (indicative of the depopulation of bilayer QWs) with time constant of $0.5 \pm 0.1$ ps. This decay time is closely matched with the fast PB building up time (indicative of being populated) at the TQWs resonance. Time-resolved PL (TRPL) also clearly shows that a fast decay of bilayer exciton PL is well-matched with a concomitant rise of the PL signal at TQWs resonances (Fig. 2a,b)—both with a lifetime limited by instrument response of 15 ps. These results therefore indicate that a substantial portion of the photo-injected excitons are localized from the bilayer QWs to the adjacent TQWs within $0.5 \pm 0.1$ ps. Following this fast exciton localization, there is a relatively slow exciton localization process from thin QWs to TQWs can be identified from both the TA and TRPL dynamics (Figs 1d and 2). The lifetime of this localization process is $\sim 200 \pm 50$ ps when the injected charge carrier density is below $10^{16}$ cm$^{-3}$ (Fig. 2b,d). We attribute this slow process to exciton localization from non-closely stacked bilayer QWs (that are located further away) to the TQWs. On increasing the injected charge carrier density, population saturation of the states in TQWs blocks the exciton localization process and the slow PB build-up is greatly accelerated (Fig. 2d).

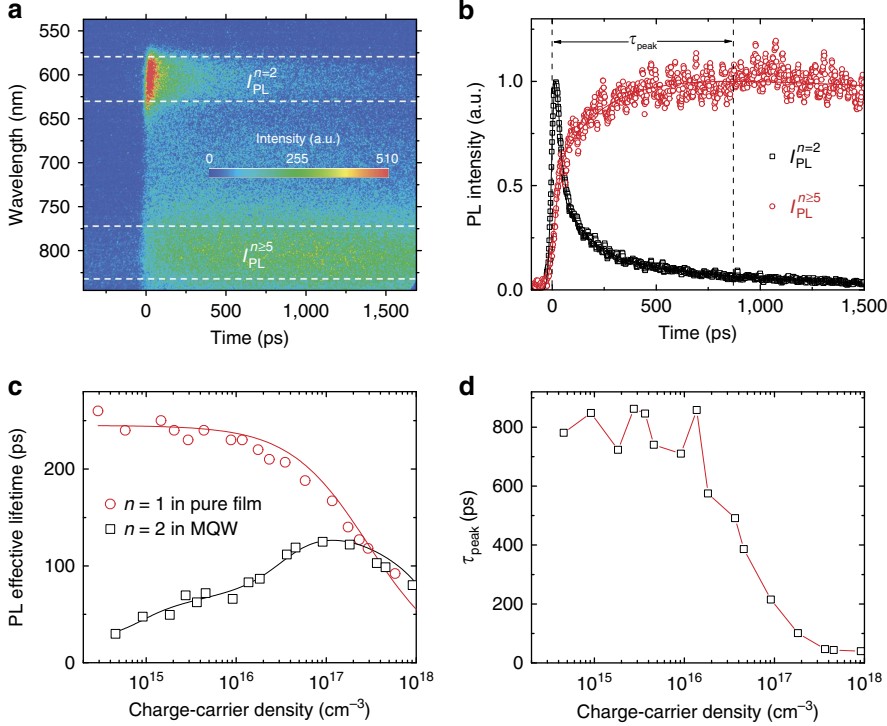

**Figure 2 | Probing exciton localization from thin quantum well (QW) to thick (T) QW using time-resolved photoluminescence (TRPL).** (**a**) Pseudo colour TRPL plot shows spectra evolution with time of the MQWs film with injected exciton density of around $1 \times 10^{16}$ cm$^{-3}$. (**b**) The normalized PL decay curves extracted at the bilayer and TQW emission position as indicated in **a**, which clearly shows the concurrence of decay at the bilayer exciton resonance and building up at the TQW exciton resonance. (**c**) A comparison of the emission effective lifetime between exciton resonance at single layer QW in its pure film and at bilayer QW in the MQWs assembles. The solid lines are fittings according to theory described in the supplementary text. (**d**) The dependence of the TQWs emission build-up time on injected carrier density in the MQW assembled film. These results were obtained following excitation at 400 nm (1 KHz, 50 fs).

Furthermore, with increasing pump intensity, more TQW states can be directly populated.

Considering that the MQWs are coupled by relatively weak van-der-Waals interactions, this observed sub-ps exciton localization time from thin QWs to closely stacked TQWs is remarkably short. In comparison, the charge transfer time is around 0.65 ps observed in epitaxially grown quasi type-II inorganic nano-heterojunctions[32], around 2 ps determined in strongly electronically coupled semiconductor NCs[33], and over 40 ps reported in van-der-Waals coupled semiconductor NCs[34]. Thus the fast exciton localization in the weakly bounded atomically flat perovskite MQWs compares favourably. A detailed understanding of this ultrafast exciton localization requires theoretical calculations to investigate the hybridization of 2D electronic states in stacked perovskite MQWs and dynamical behaviour of photoexcited states induced by electron-electron and electron-phonon couplings. The quantum coupling effect, which is proposed for ultrafast hole transfer in atomically thin type-II MoS$_2$/WS$_2$ heterostructures[25], may also be at work here. This could be validated using 2D coherence spectroscopy, but this is beyond the scope of our work here. Nonetheless, the ultrafast exciton localization and relatively weak van-der-Waals coupling between stacked QWs have important implications for LED application. They enable perovskite MQWs the means to transcend the traditional conflicting constraints of high charge carrier injection rate and effective electronic isolation (high-luminescence QY) of the emissive center in NC-thin-film LEDs, and the extremely challenging requirements of hetero-epitaxially hybridizing the NCs with a large-bandgap conductive host without interfacial trap states in NC-doped LEDs.

The observed sub-ps exciton localization from the closely stacked thin QWs to TQWs is much shorter than the exciton lifetime and most other dynamical processes in thin QWs, which are on the order of several tens to a few thousand picoseconds. The relatively slow exciton localization (around 200 ps) between the non-closely stacked QWs can also effectively compete with the other dynamical processes. This effective competition manifests as a significantly shortened residual bilayer PL lifetime of the MQWs compared to that of pure single-layer QWs (Fig. 2c). In contrast, the excitonic PL lifetime should increase when increasing QW thickness due to the reduced oscillator strength[21–23]. These fast exciton localization processes indicate that the overall exciton transfer efficiency from thin QWs to TQWs should be extremely high, which can be sensitively probed with steady-state PL spectroscopy. The exciton localization will shift PL from thin QWs resonance energy to TQWs resonance energy. Figure 1c and Supplementary Fig. 11 clearly show that over 95% of PL from the MQWs is dominated by the recombination in TQWs, while the excitons are primarily injected into bilayers with density smaller than $10^{16}$ cm$^{-3}$. When the injected carrier density is increased above $10^{16}$ cm$^{-3}$, the electronic states in TQWs will be partially saturated, which blocks further exciton localization from thin QWs to TQWs. Therefore, two high energy PL peaks from the bilayer and trilayer QWs emerge. The PL peak intensity ratios of bilayer and trilayer to TQWs grow super linearly with the injected carrier density. Such carrier density dependent PL ratio is a clear signature of the efficient exciton transfer from thin QWs to TQWs[35]. Next, a carrier density dependent PL emission model based on exciton localization was employed to estimate the exciton transfer

efficiency (Supplementary Note 3). When the injected exciton density is below $10^{16}$ cm$^{-3}$, the exciton transfer efficiency is estimated to be over 85% (Supplementary Fig. 11d). Remarkably, the effective emission colour of the van-der-Waals coupled MQWs can be largely tuned with the injected carrier density (Fig. 1c), suggesting that we could also achieve emission colour tunability with weakly bound 2D stackers, and is not limited to only epitaxially grown heterojunctions[32,35].

**Fast and efficient excitonic recombination in perovskite MQWs.** In lead-based 3D perovskites, the second order bimolecular recombination rate is proportional to the free electron concentration multiplied by free hole concentration. Hence, decreasing the injected carrier density will result in a quadratic power dependent decrease of the recombination rate. Under LED working conditions, where the carrier density typically is $<10^{15}$ cm$^{-3}$, the lifetime for electrons and holes to recombine with each other is longer than 10 μs (considering that the bimolecular recombination rate as $1 \times 10^{-10}$ cm$^3$ s$^{-1}$—refer to latter discussion) after they were injected into the same space. Such long dwell time leads to charge carrier recombination at even relatively low defect states densities (first order) as the charges wander or pass through the active layer without recombination. Therefore, the relatively slow bimolecular recombination in these 3D perovskites is a fundamental limiting factor in the development of high-efficiency LEDs. Earlier, we have shown that the injected charge carrier localization from thin QWs to TQWs is extremely fast and efficient, though they are coupled with relatively weak van-der-Waals interaction. Next, we further demonstrate that light emission originates from excitonic recombination in the TQWs with a much larger decay rate and higher efficiency than the bimolecular recombination in the 3D counterparts.

With 1.9 eV pulsed excitation, excitons were directly injected into the TQWs in the MQW assemblies. Under low-fluence pumping, Fig. 3a shows that the PL effective lifetime ($\tau_{PL}^{1/e}$) of the MQWs (around 20 ns) is nearly two orders of magnitude shorter than that of the 3D counterpart (over 1 μs; Table 1), which indicates that the luminescence origins in these two systems should be different from each other. Figure 3b,c summarize the initial time PL intensity ($I_{PL}[t=0]$) and $\tau_{PL}^{1/e}$ as a function of injected carrier density for both the MQWs and 3D perovskite, respectively. The quadratic behaviour for $I_{PL}[t=0]$, which gives a clear signature of the quadratic power dependence of the recombination rate on the carrier densities, confirms the bimolecular type recombination in 3D perovskite. Accordingly, the $\tau_{PL}^{1/e}$ of 3D perovskite continually decreases with increasing injected carrier density (Fig. 3c). The detailed dependence of $\tau_{PL}^{1/e}$ on carrier density can be well-modeled with bimolecular recombination (second order dependence) coupled with charge carrier trapping (first-order dependence) and three particle Auger recombination (third order dependence; Supplementary Note 4). Therefore, the luminescence QY should first increase with increasing carrier density due to the gradual domination of the bimolecular recombination over the charge carrier trapping. At high carrier densities, where the three particle Auger recombination becomes effective and dominates over the bimolecular recombination, the luminescence QY then decreases with increasing carrier density. Such a peak shape dependence for 3D perovskite is unambiguously revealed in Fig. 3d. The peak luminescence QY is a compromise between charge carrier trapping, bimolecular recombination and Auger recombination. Given that the bimolecular recombination coefficient and Auger recombination coefficient are intrinsic to the 3D perovskite, a peak luminescence QY as high as 50% is achieved by engineering the trap states density down to a low level—$3.5 \times 10^{16}$ cm$^{-3}$ (Fig. 3d and Supplementary Fig. 15).

On the other hand, for perovskite MQWs, $I_{PL}[t=0]$ is linearly dependent on the injected carrier density and luminescence possesses a near invariant $\tau_{PL}^{1/e}$ at carrier densities below $1.5 \times 10^{16}$ cm$^{-3}$ (Fig. 3b,c). These results suggest that luminescence in TQWs (following fast exciton localization) originates from first-order excitonic emission or trap-assisted monomolecular recombination under low carrier density condition. The later process will result in relatively low luminescence QY and the QY increases with increasing the injected carrier density due to saturation of the trap states. However, the luminescence QY as high as 60% and near invariant dependence have been observed for MQWs at carrier density below $10^{16}$ cm$^{-3}$. Above $10^{16}$ cm$^{-3}$, the QY significantly decreases with increasing carrier densities (Fig. 3d). These facts validate that the luminescence in TQWs is from excitonic recombination rather than trap-assisted recombination. The trap density associated with TQWs is determined to be ultra-low (lower than $3 \times 10^{13}$ cm$^{-3}$, Table 1, Supplementary Fig. 15 and Supplementary Note 5). The relatively weak van-der-Waals coupling between adjacent QWs render the TQWs electronically isolated and passivated. Under the same injected carrier density, the $\tau_{PL}^{1/e}$ of this excitonic recombination in TQWs is one to two orders of magnitude faster than that of the bimolecular recombination in 3D perovskites (Fig. 3c and Table 1). The nearly invariant high luminescence QY (60%) originates from efficient charge localization, fast excitonic recombination and low trap states density associated with the TQWs. Further increase of the injected carrier density to above $1.5 \times 10^{16}$ cm$^{-3}$, $I_{PL}[t=0]$ shows a transition from linear to super-linear dependence, which reveals the increased contribution from free electron-hole bimolecular recombination. This result suggests that excitons tend to dissociate to free charge carriers under multi-particle interaction and charge screening effects at high density. By fitting the dependence of $I_{PL}[t=0]$ on carrier density (considering Poisson-distributed multi-exciton dissociation) in TQWs[36], the self-assembled TQWs concentration in the assembly is estimated to be around $9 \times 10^{16}$ cm$^{-3}$ (Supplementary Note 6). Accompanying the multi-particle induced exciton dissociation, non-radiative Auger processes also becomes obvious in TQWs. Therefore, the decreasing $\tau_{PL}^{1/e}$ at high carrier density can be well-described with exciton recombination coupled with bimolecular recombination and three-particle Auger recombination (Fig. 3c). When the charges are primarily injected into bilayer QWs, in addition to Auger recombination, the blocking-induced exciton recombination in thin QWs should also be responsible for the decreased luminescence QY observed in TQWs at high carrier density (Fig. 3d). Nevertheless, the MQWs show a nearly invariant high QY excitonic recombination at carrier density below $1.5 \times 10^{16}$ cm$^{-3}$. In contrast, the luminescence QY of the bimolecular recombination in 3D perovskite significantly decreases with decreasing injected carrier densities (Fig. 3d). The absolute luminescence QY value may slightly vary with the chemical composition of the perovskite. However, the dependence of the luminescence QY on carrier density should persist since it is governed by the basic carrier decay physics, which is determined by the structural dimension of the perovskite, regardless of the composition. Our findings clearly demonstrate that the slow bimolecular recombination limitation in 3D lead-halide perovskites for EL at relative low carrier density (smaller than $10^{15}$ cm$^{-3}$) can be effectively overcome by using these self-assembled perovskite MQWs.

## Discussion

In 3D lead-halide perovskites (including $CH_3NH_3PbI_3$ (MAPbI$_3$), FAPbI$_3$, MAPbI$_{3-x}$Cl$_x$, FAPbI$_{3-x}$Cl$_x$ and so on), the primary photo-generated excitons will quickly dissociate to free electrons

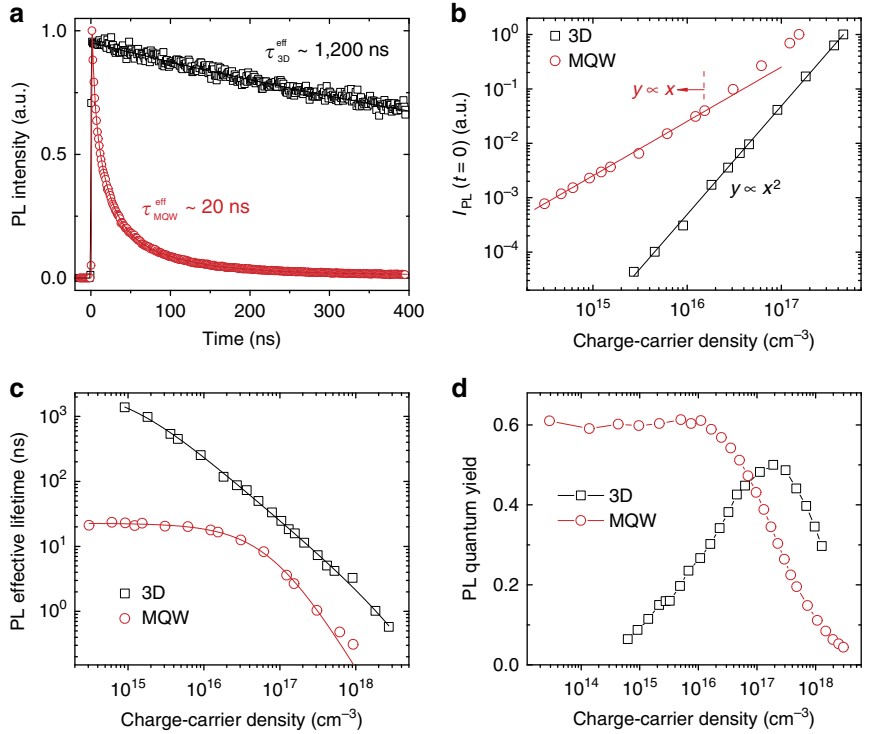

**Figure 3 | Bimolecular emission in three-dimensional (3D) perovskites versus excitonic emission in perovskite multi-quantum wells (MQWs)**
(**a**) Time-resolved photoluminescence (TRPL) decay transients measured at $812 \pm 60$ nm for 3D perovskite and $782 \pm 60$ nm for perovskite MQW films following excitation at 650 nm (1 KHz, 100 fs, around $0.01\,\mu J\,cm^{-2}$). The solid lines are exponential fits of the PL decay transients. (**b**) Photon-injected carrier density dependence of the initial time PL intensity ($I_{PL}(0)$) following excitation at 650 nm (1 KHz, 100 fs). The quadratic dependence indicates that free-carrier bimolecular recombination dominates charge carrier decay in 3D perovskite, while the linear dependence suggests excitonic recombination dominates this high PLQY charge carrier decay in perovskite MQWs. (**c**) The plots of PL effective lifetimes as a function of photon-injected carrier density following excitation at 650 nm (1 KHz, 100 fs). The solid lines are fittings according to theory described in the supplementary text. (**d**) PL quantum yield as a function of the photon-injected carrier density of the films measured with 405 nm laser pulses (2.5 MHz, 200 ps).

**Table 1 | The PL decay effective lifetime ($\tau_{PL}^{1/e}$) and quantum yield ($\eta_{PL}$) at injected carrier density of around $10^{15}$ cm$^{-3}$, Trap states density ($\eta_{Trap}$), First order ($k_1$), Second order ($k_2$), and Third order (three-particle Auger) recombination constant ($k_3$), Biexciton Auger recombination coefficient ($C_{bx}$).**

| | $\tau_{PL}^{1/e}$(ns) | $\eta_{PL}$ | $\eta_{Trap}$ (cm$^{-3}$) | $k_1$ (s$^{-1}$) | $k_2$ (cm$^3$ s$^{-1}$) | $k_3$ (cm$^6$ s$^{-1}$) | $C_{bx}$ (cm$^2$ s$^{-1}$) |
|---|---|---|---|---|---|---|---|
| 3D | $1,300 \pm 100$ | $0.08 \pm 0.02$ | $3.5(\pm 0.8) \times 10^{16}$ | $9(\pm 4) \times 10^5$ (Trap-assisted) | $7(\pm 3) \times 10^{-10}$ | $3(\pm 2) \times 10^{-28}$ | — |
| MQWs | $22 \pm 2$ | $0.60 \pm 0.05$ | Lower than $3 \times 10^{13}$* | $5(\pm 3) \times 10^7$ (Excitonic) | $2(\pm 0.6) \times 10^{-9}$ | $2(\pm 1) \times 10^{-26}$ | — |
| 2D ($n=2$) | —† | —† | —† | —† | — | — | $0.10 \pm 0.02$ |
| 2D ($n=1$) | $0.25 \pm 0.03$ | $0.02 \pm 0.01$ | —‡ | $4.0(\pm 0.4) \times 10^9$ (Both) | — | — | $0.44 \pm 0.09$ |

TQW, thicker quantum well; 2D, two dimensional; 3D, three dimensional.
*The trap states density associated with the TQWs ($n \geq 5$) in the MQWs film.
†We could not achieve pure quasi-2D ($n=2$) perovskite bulk crystal film.
‡2D ($n=1$) perovskite is not stable under high-power laser excitation.

and holes. This fact has been confirmed with various ultrafast spectroscopy techniques[8,37–40]. The subsequent free electron-hole bimolecular recombination rate is nearly four orders of magnitude smaller than that expected from Langevin theory (which assumes the free electron and hole will recombine once they meet each other within their capture radius)[8]. This slow bimolecular recombination provides a relatively long time for the photoexcited electrons and holes to be efficiently extracted out from the active layer, which is one of the main merits that drives perovskite's superior light collecting performance[8,37–40]. However, for light emitting diodes application, this is a fundamental limitation. The charge carrier kinetics in 3D lead based perovskites can be described with the following differential equation[8,37–40].

$$\frac{dn(t)}{dt} = G - k_1 n - k_2 n^2 - k_3 n^3 \qquad (1)$$

where $n$ is the charge carrier density and $t$ is time, $G$ is generation rate of the charge density, $k_1$ is the trap-mediated monomolecular recombination constant, $k_2$ is the free carrier bimolecular recombination constant and $k_3$ is the three-body Auger recombination constant. Under steady-state electrical injection, the radiative emission quantum yield ($\eta(n)$) is given by

$$\eta(n) = \frac{nk_2}{k_1 + nk_2 + n^2 k_3} \qquad (2)$$

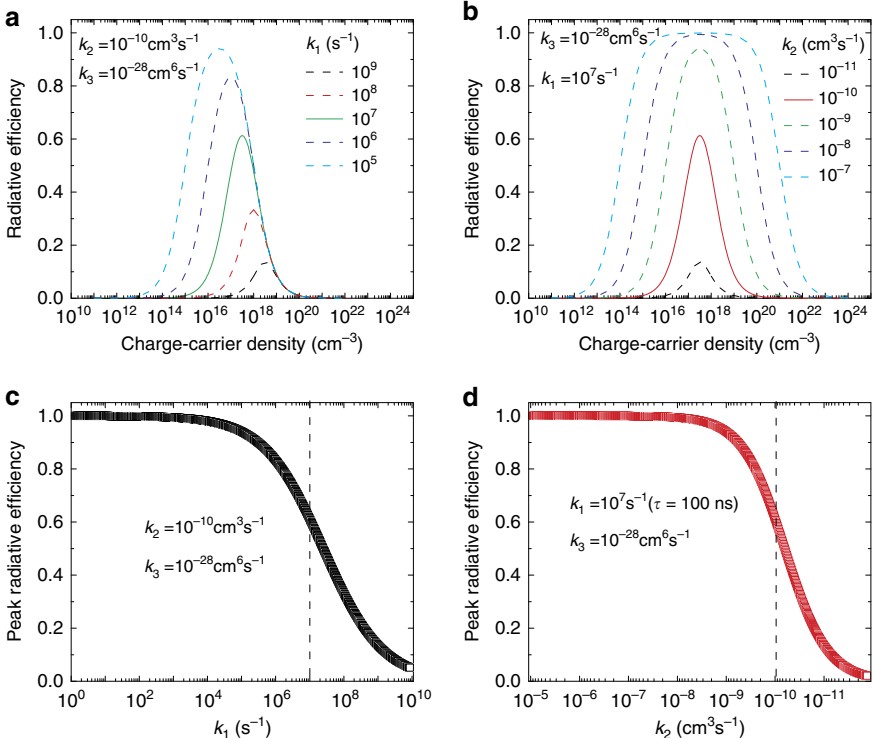

**Figure 4 | The calculated luminescence quantum yield (QY) in three-dimensional (3D) lead-based perovskites.** (**a**) Injected carrier density dependent luminescence QY with fixed reported bimolecular recombination constant ($k_2$) and Auger recombination constant ($k_3$), with several representative trap mediated recombination constants ($k_1$). (**c**) With fixed $k_2$ and $k_3$, the dependence of highest achievable luminescence QY on $k_1$. The dashed line indicates the typical reported $k_1$ value ($10^7 \text{s}^{-1}$, 100 ns). (**b**) Injected carrier density dependent luminescence QY with fixed $k_1$ and $k_3$, but with different $k_2$. (**d**) With fixed $k_1$ and $k_3$, the dependence of highest achievable luminescence QY on $k_2$. The dashed line indicates the typical reported $k_2$ value ($10^{-10} \text{cm}^3 \text{s}^{-1}$).

Therefore, the luminescence quantum yield should be strongly dependent on $n$. It first increases with increasing $n$ due to the gradual domination of the second order bimolecular recombination over the first order charge carrier trapping. At high carrier densities, where the three particle Auger recombination becomes effective and dominates over the bimolecular recombination, the luminescence quantum yield will in turn decrease with increasing $n$. The peak luminescence quantum yield $((1 + 2(k_1 k_3)^{0.5} k_2^{-1})^{-1})$ is reached at the carrier density of $k_1^{0.5} k_3^{-0.5}$. The calculated dependence of luminescence quantum yield on $n$ based on equation (2) clearly shows the peak shape dependence (Fig. 4), which is closely matched with the experimentally observed dependence (Fig. 3d). This concurrence indicates that equation (2) works well for 3D lead-based perovskites. Since $k_2$ (around $10^{-10} \text{cm}^3 \text{s}^{-1}$) and $k_3$ (around $10^{-28} \text{cm}^6 \text{s}^{-1}$) are the intrinsic parameters of the 3D lead-halide perovskite[8,37–40], the luminescence quantum yield achieved is strongly dependent on $k_1$ and $n$. To achieve a 50% internal luminescence quantum yield, even driving the light emitting diode at a high carrier density of $10^{15} \text{cm}^{-3}$, the trap mediated recombination lifetime must be engineered to be as long as the electron-hole bimolecular recombination lifetime—10 µs (Fig. 4a, $k_1 = 10^5 \text{s}^{-1}$). Preparing 3D perovskite thin films with such long trap related recombination lifetime (low trap states density) is extremely challenging. Up to now, the longest reported trap mediated recombination lifetime is only few µs (refs 41). During such long dwell time (10 µs), given the high charge carrier mobility reported in these perovskites[7], the charge carriers also have a very high possibility to pass through the thin active layer without recombination. If the light emitting diode is driven at a lower carrier density or with a higher trap states density

(trap-related recombination time shorter than 10 µs), the trap related recombination will always dominate over the free electron-hole bimolecular recombination, which will result in an internal quantum yield lower than 50% (Fig. 4). Therefore, the slow bimolecular recombination in 3D lead-based perovskite is a fundamental limitation for application in LED under relatively low concentration of injected carriers. For coherent light emission in these perovskites, the injected carrier density is typically higher than $10^{17} \text{cm}^{-3}$. Under this condition, the electron-hole bimolecular recombination and subsequent avalanche can effectively compete with the charge carrier trapping[6,38].

One possible way to achieve high luminescence quantum yield at $n \leq 10^{15} \text{cm}^{-3}$ and moderate trap states density ($k_1 \sim 10^7 \text{s}^{-1}$) is by replacing lead-based perovskite with other types of perovskites which has higher bimolecular recombination constants[42]. On the basis of calculations shown in Fig. 4c, LEDs driven at carrier density of $10^{15} \text{cm}^{-3}$, 50% internal luminescence quantum yield could be still be achieved if we could increase $k_2$ to $10^{-8} \text{cm}^3 \text{s}^{-1}$ (that is, two orders of magnitude larger than the lead counterparts); while maintaining $k_1$ at $10^7 \text{s}^{-1}$. However, such perovskite with these properties have yet to be realized.

For the pure 2D case, the direct-bandgap lead-halide single layer 2D ($n=1$) perovskites are monomolecular layers of [PbI₆] octahedral sandwiched between long organic barrier layers. The band gap of the barrier layer is much larger than that of the lead-halide inorganic well layer (by at least 3 eV), and the dielectric constant of the barrier layer (around 2.1) is much smaller than that (around 6.1) of the well layer. As a result, the excitons are tightly confined by these two effects within the well[18–23,26,27]. Unlike 3D perovskites, the termination of the crystal will result in

dangling bonds. An ideal lead-halide 2D perovskite has natural out-of-plane self-termination without any dangling bonds. Therefore, near 100% luminescence quantum yield is expected. However, due to the presence of defects and some other unknown reasons, the quantum yield in lead-halide 2D ($n = 1$) perovskite bulk crystals has typically been very poor[18–23,26,27]. Efficiencies of the light emitting diodes constructed with monomolecular layer 2D perovskite are extremely low and typically measured at cryogenic temperatures[18,19]. Here, through detailed pump fluence and temperature dependent ultrafast spectroscopy investigation of the 2D $(NMA)_2PbI_4$ perovskite, we show that the injected carriers decay through significant surface trapping, intrinsic defect trapping, exciton-phonon coupling and exciton-exciton interaction. These decay processes can effectively compete with the band edge radiative recombination and are responsible for the low luminescence quantum yield observed (see Supplementary Note 2 for details).

With efficient and fast excitonic recombination (first order) in the thick (inorganic layer number $n \geq 5$) quantum wells, the van-der-Waals coupled perovskite multi-quantum well assemblies transcend the slow bimolecular recombination bottleneck in 3D lead-halide perovskites for electroluminescence. Furthermore, the multi-quantum well assemblies bypass the significant non-radiative recombination faced by the pure 2D perovskites with quantum coupling-induced ultrafast exciton localization from thin ($n \leq 4$) quantum wells to thick ($n \geq 5$) quantum wells. The relatively weak van-der-Waals coupling between adjacent quantum wells affords the thick ($n \geq 5$) quantum wells electronical isolation and passivation. The self-assembled thick ($n \geq 5$) quantum wells inherit the high crystallinity and low trap density of perovskite nanocrystals. Our findings reveal an exciton transfer efficiency larger than 85% from the thin quantum wells to thick quantum wells. In light-emitting devices, electrons and holes are separately electrically injected into the multi-quantum well, the probability of exciton recombination within the thin quantum wells during the dynamical localization process is greatly reduced. Therefore, the carrier localization efficiency from thin quantum wells to thick quantum wells under electrical luminescence should be even higher than that measured with the photoluminescence approach presented here. In retrospect, the Ruddlesden-Popper phase perovskite multi-quantum well assemblies provide a viable approach to realize new functionalities that are not available in single phase constituents—through facile van-der-Waals coupled 2D layers rather than epitaxial-grown heterojunctions. Importantly, our findings would help accelerate the low-temperature solution-processed perovskite electroluminescence technology closer to meeting the commercial requirements for low-cost, large-area lighting sources and displays.

## Methods

**Materials preparation.** To deposit the perovskite MQWs films for optical characterization, the precursors of $C_{10}H_7CH_2NH_3I$ (NMAI), $CH_5N_2$ (FAI) and $PbI_2$ with a molar ratio of 2:1:2 were dissolved in DMF (40 wt.%). The obtained solution was stirred at 60 °C for 2 h in a nitrogen-filled glovebox. Then the films were prepared by spin coating the solution onto quartz substrates at 2,000 r.p.m., followed by annealing on a hot plate at 100 °C for 10 min. The 2D $(NMA)_2PbI_4$ and 3D $FAPbI_3$ films were prepared with the same methods described above by adding stoichiometric precursors in DMF (40 wt% for 2D, 20 wt% for 3D). All optical measurements were conducted in an optical cryostat under nitrogen-filled or vacuum.

The precursor NMAI was synthesized by adding 4.34 g hydroiodic acid (45 wt% in water) to a stirring solution of 1-naphthalenemethylamine (12.72 mmol) in tetrahydrofuran (THF, 50 ml) at 0 °C for 2 h. The solution was then evaporated at 50 °C to obtain the NMAI precipitate, which was washed three times with THF:$CH_2Cl_2$ (3:1) mixture and then dried under vacuum. The other precursors were bought from Sigma without any further purification.

**Characterization.** The scanning electron microscopic images of the perovskite films were obtained with a scanning electron microscope (JSM-6700F) operated at

3 KeV. The perovskite films were sputtered with a thin layer of gold at 20A and 30 s. The X-ray diffraction spectra were recorded with Bruker D8 Advance. The thickness of the studied films was determined with the Alpha-Step IQ stylus-based surface profiler measurements.

**Absorption and photoluminescence quantum yield.** Ultraviolet-vis absorbance spectra were recorded on a ultraviolet-Vis spectrophotometer (UV-1750, SHIMADZU). Room temperature relative PLQY at different excitation intensity of the perovskite thin films was measured with a time-resolved confocal microscope (MicroTime 200, PicoQuant). The samples were excited with 405 nm (200 ps, 2.5 MHz, less than 3 mW) laser pulses generated from the LDH-P-C-405B laser diode (PicoQuant). The luminescence was then passed through a dichroic mirror (T425lpxr, Chroma) and an 715 nm long pass filter before being detected by a single-photon avalanche diode (SPAD, SPCM-AQR-15, Perkin-Elmer). The PL at different excitation intensity was recorded under same exact experimental conditions except for the excitation intensity, which was controlled by adjusting the neutral density filters. Under the assumption of negligible nonlinear absorption and negligible sample damage, the relative PLQY at different excitation intensity was obtained by normalizing the collected PL intensity with the corresponding excitation light intensity. The relative PLQY at high carrier density (higher than $10^{17} cm^3$) may be slightly underestimated due to the small saturable absorption of the samples at 405 nm. A three-step technique was used to obtain the absolute PLQY values of perovskite films by combination of a 445 nm continuous wave (CW) laser, optical fiber, spectrometer and integrating sphere[43].

**Photoluminescence and transient photoluminescence.** For femtosecond optical spectroscopy, the laser source was a Coherent Libra regenerative amplifier (50 fs, 1 KHz, 800 nm) seeded by a Coherent Vitesse oscillator (50 fs, 80 MHz). 800 nm wavelength laser pulses were from the regenerative amplifier while 400 nm wavelength laser pulses were obtained with a BBO doubling crystal. 650-nm laser pulses were generated from a Coherent OPerA-Solo optical parametric amplifier. The laser pulses (circular spot, diameter 2 mm) were directed to the films. The emission from the samples was collected at a backscattering angle of 150° by a pair of lenses into an optical fiber that was coupled to a spectrometer (Acton, Spectra Pro 2500i) and detected by a charge coupled device (Princeton Instruments, Pixis 400B). TRPL was collected using an Optronis Optoscope streak camera system which has an ultimate temporal resolution of around 10 ps.

**Transient absorption.** The broadband femtosecond TA spectra of the perovskite films were taken using the Ultrafast System HELIOS TA spectrometer. The laser source was the Coherent Legend regenerative amplifier (150 fs, 1 KHz, 800 nm) seeded by a Coherent Vitesse oscillator (100 fs, 80 MHz). The broadband probe pulses (420–800 nm) were generated by focusing a small portion (around 10 μJ) of the fundamental 800 nm laser pulses into a 2 mm sapphire plate. The 400-nm pump pulses were obtained through doubling the fundamental 800 nm pulses with a BBO crystal. 650-nm pump pulses were generated from a Light Conversion TOPAS-C optical parametric amplifier.

**Data availability.** The experimental data that support the findings of this study are available from the corresponding author on request.

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

## Acknowledgements

We thank Dr Nana Wang and Dr Jianpu Wang from Nanjing Tech University (Nanjing Tech), China, for providing the high-quality samples. Financial support from Young 1000 Talents Global Recruitment Program of China, the National Basic Research Program of China- Fundamental Studies of Perovskite Solar Cells (2015CB932200), the National Natural Science Foundation of China (61605073), Nanyang Technological University start-up grant M4080514, the Ministry of Education Academic Research Fund Tier 1 grant RG101/15 and Tier 2 grants MOE2013-T2-1-081, MOE2014-T2-1-044 and MOE2015-T2-2-015; and grants from the Singapore National Research Foundation through the Singapore–Berkeley Research Initiative for Sustainable Energy (SinBeRISE) CREATE Program and the Competitive Research Program NRF-CRP14-2014-03 is gratefully acknowledged.

## Author contributions

G.X., T.C.S. and W.H. conceived the idea for the manuscript and designed the experiments. G.X. developed the basic concepts, conducted the spectroscopic characterization and coordinated the experiments. All authors co-wrote the manuscript. T.C.S. and W.H. led the project.

## Additional information

**Competing financial interests:** The authors declare no competing financial interests.

**Publisher's note**: 

