## [Peer Review File · Nature Communications]

Reviewers' comments:

Reviewer #1 (Remarks to the Author):

In this manuscript, the authors prepared three-dimensional (3D) perovskite crystals and two-dimensional (2D) quantum-well perovskite crystals and studied their photoluminescence (PL) properties. They reported the enhancement of the PL intensity and the short PL lifetime in 2D crystals: The exciton plays an essential role in the radiative processes in 2D crystals. However, these observation and conclusions have been already reported in the early papers in the 1990s. In addition, there is no experimental demonstration of the EL device fabrication and operation, although the title of the present manuscript is "for electroluminescence". The electroluminescence devices based 2D crystals have been also demonstrated (e.g., ref. 12). Then, the scientific merits do not match the high quality requirements of Nature Communications. The discussions and conclusions in this manuscript are not significant enough to warrant publication in Nature Communications.

Reviewer #2 (Remarks to the Author):

In the current manuscript the authors describe how to effectively overcome the low bimolecular recombination rate in perovskites by using 2-D, van-der-Waals assembled perovskite building blocks which results in favorable recombination when compared with 3-D bulk perovskites. The manuscript is generally free of errors except for a lonely ")" in the abstract (line 21) and an "additional" on page 9, line 11. The references are adequate with a few exceptions as noted below.

The methods applied, the data gathered and the analysis performed are excellent. The findings are novel and provide a clear strategy into further improvement of light-emitting diodes incorporating this emitter materials. Therefore, the reviewer strongly recommends publication of this manuscript with very minor revisions.

1. The only concern with the manuscript at the current stage is that it is very hard to extract the numbers that align with the title of the manuscript, e.g. how did the respective recombination rates change when move from 3D to 2D quantum wells; this information should be made accessible more easily, in a table for example as it is hard to extract from Figure 3d and the supporting information alone.

2. With respect of the diffusion length of excitons in this materials I would recommend the authors to Brenner et al. (J. Phys. Chem. Lett., 2015, 6 (23), pp 4754-4757) and with respect to the electron-phonon coupling to Wright et al. (Nature Communications 7, Article number: 0 (2016) doi:10.1038/ncomms11755).

3. The authors missed a relevant reference related to early stage perovskite LEDs. The reviewer recommends to cite the following refernece in the introduction.

[1] Y.-H. Kim et al, Adv. Mater., 27, 1248-1258 (2015)

Reviewer #4 (Remarks to the Author):

In this paper, Xing et al elucidated that van-der-Waals coupled perovskite multiple quantum-wells (MQWs) exhibit much faster and thus much more efficient radiative excitonic recombination, compared with conventional 3D perovskites or 2D perovskites. Such MQWs intrinsically overcome the bottleneck of the slow bimolecular recombination in lead-halide perovskites which limits the electroluminescent efficiency; very promising for high efficient electroluminescent applications.

Using ultrafast time-resolved PL and transient techniques, they confirmed the very rapid localization (~ 0.5 ps) occurring from adjacent thin few layer ($n \leq 4$) MQW to the thick ($n \geq 5$) MQW with extremely high efficiency ($>85\%$), avoiding conventional large defect trapping. The experimental results well support their conclusion and provide novel insight into the carrier dynamics in the perovskite MQWs. The manuscript is well organized and presented. This manuscript may be publishable if authors can address the following concerns and improve the manuscript:

1. The most conclusions are based on the comparison between MQWs and 2D (NMA)₂PbI₄ and 3D FAPbI₃ films. It is well known that compositions of perovskite significantly impact the carrier dynamics, in addition to structures. It is essentially important to justify the validity of the comparison;
2. It is not evident that the perovskite MQWs remain high charge carrier mobility of halide perovskite. So it is not a suitable conclusion, particularly in abstract;
3. When discussing 2D perovskites, the statement originates from the literatures published long time ago. Due to recent significant progress (for example, Dou et al in Science and Tsai et al in Nature), the statement may not really suitable.
4. Line 176 "Figures 1c & S12 clearly show that over 95% of PL from the MQWs is dominated by the recombination in TQWs while the excitons are primarily injected into bilayers with density $<10^{16}$ cm⁻³". The 95% is not clearly shown and inconsistent to FigS12d.
5. Line 293 "Room temperature relative PLQY at different excitation intensity of the perovskite ..."
Here PLQY is incorrect because you actually do not take into account the different absorption and scattering.
6. Fig S19a may be not valid, given very low intensity.

Reviewer's comments to the author

Reviewer 1

In this manuscript, the authors prepared three-dimensional (3D) perovskite crystals and two-dimensional (2D) quantum-well perovskite crystals and studied their photoluminescence (PL) properties. They reported the enhancement of the PL intensity and the short PL lifetime in 2D crystals: The exciton plays an essential role in the radiative processes in 2D crystals. However, these observation and conclusions have been already reported in the early papers in the 1990s. In addition, there is no experimental demonstration of the EL device fabrication and operation, although the title of the present manuscript is “for electroluminescence”. The electroluminescence devices based 2D crystals have been also demonstrated (e.g., ref. 12). Then, the scientific merits do not match the high quality requirements of Nature Communications. The discussions and conclusions in this manuscript are not significant enough to warrant publication in Nature Communications.

Firstly, we would like to thank the reviewer for his time and the comments regarding our manuscript. Here, we need to make a clear and consistent definition of the lead-halide perovskite dimensionality. Traditional 2D ($n = 1$) and quasi-2D ($n \geq 2$) perovskite quantum-well (QW) means n layers of inorganic $[\text{MX}_6]$ octahedral monomolecular sheets are sandwiched by two long-chain organic barrier layers. The exciton binding energy and oscillator strength are increasing with decreasing the inorganic layer number (n). We fully agree with the referee that these 2D perovskites have been reported in 1990s. We have also clearly stated these arguments and cited the relevant references. For example in the original manuscript, lines 76-79:

“The excitons are quantum confined within the inorganic wells, but the quantum confinement effect decreases with increasing well width. Hence, the excitonic properties of the MQW can be systematically tailored from 2D to 3D by increasing the inorganic layer number from $n = 1$ to $n = \infty$ ¹⁸⁻²⁰.”

However, the material system investigated in our work is the self-assembled multi-QWs (MQWs) crystal film (a combination of $n = 2, 3, 4, 5 \dots$ MQWs in one crystal film). It contrasts with the traditional pure 2D or quasi-2D QWs crystal film (with fixed n in the film). Such self-assembled MQWs have never been investigated in the 1990s. Hence, the reviewer's comment that “*The enhancement of the PL intensity and the short PL lifetime in 2D crystals: The exciton plays an essential role in the radiative processes in 2D crystals*” as a summary of the work in our manuscript is therefore inaccurate. In fact, we had clearly revealed that the PL

intensity (efficiency) of the 2D perovskite ($n=1$, with the largest quantum confinement and largest exciton binding energy among the 2D perovskites) bulk crystal is extremely low, due to the 2D perovskite bulk crystals suffering from significant non-radiative recombination (*i.e.*, surface trapping, intrinsic defect trapping, extremely strong exciton-phonon coupling and exciton-exciton interaction) (Please refer to part 3 of the supplementary information (SI) for the details). Furthermore, we never conclude that the PL intensity (or efficiency) of the 2D perovskite bulk crystal is larger than that of the 3D counterpart. Though the 3D lead-halide perovskite suffers from the slow free electron-hole bimolecular recombination. In our manuscript, we clearly state that both the 3D (Original manuscript, lines 45-52) and traditional 2D (Original manuscript, lines 59-64) perovskites are not ideal for electroluminescence. We respectfully disagree with the referee's comment that "... *these observation and conclusions have been already reported in the early papers in the 1990s.*"

Most recently, remarkably high efficiency perovskite light emitting diodes (LEDs) constructed with the MQW assemblies have been demonstrated (refs. 14, 15). However, the innate dynamics of the injected charge carriers in these MQWs assemblies driving the high efficiencies in these LEDs remain unclear. We would like to clarify that in this manuscript, we advance the fundamental photophysical knowledge into overcoming the intrinsic limitations (as stated above) of 2D and 3D perovskites and towards achieving high efficiency perovskite LEDs. In this work, we clearly rationalize how these limitations in 2D and 3D perovskites are overcome with the Van-der-Waals coupled 2D-3D perovskite mixtures (self-assembled MQWs) through detailed femtosecond transient optical spectroscopy. The MQWs bypass the significant non-radiative recombination faced by the 2D perovskite with quantum coupling-induced ultrafast charge localization from thin QWs (inorganic layer number $n \leq 4$) to thick (T) QWs ($n \geq 5$). The relatively weak van-der-Waals coupling between adjacent QWs rendered the TQWs electronically isolated and passivated. The self-assembled TQWs inherit the high crystallinity and low trap density of perovskite nanocrystals. The fast and efficient excitonic recombination in the TQWs allow the MQW assemblies to effectively overcome the slow bimolecular recombination bottleneck for 3D perovskite LEDs. These complementary properties in van-der-Waals coupled MQWs which are typical of epitaxial vacuum-grown heterojunctions, render us new functionalities that are not available with traditional halide perovskite single phase constituents. More specifically the key points from our study are:

1. *First report on the clear photophysical picture of the fundamental limitations in the 3D and 2D lead-halide perovskites for LEDs.*

2. *First report presenting concrete evidences (experimental and modelling) on how the limitations in 3D and 2D perovskites are overcome by using the perovskite MQWs.* Although high efficiency perovskite LEDs constructed with the MQW assemblies have been demonstrated, the innate dynamics of the injected charge species (free-carrier or excitonic recombination) in these MQWs assemblies driving the high efficiencies in these LEDs remain unknown.
3. *First report on how to achieve the emission color tunability (dual emission) with weakly bound 2D perovskite stackers as well as detailed insights into QW-width dependent charge carrier dynamics.*

We hope that our above justifications could satisfy the referee concerns. For better clarity, we also made the following changes:

In original manuscript, lines 72-82:

“In-between these 2D and 3D classes of perovskites, there is also a family of perovskite multi-quantum-well (MQW) crystals $L_2(\text{SMX}_3)_{n-1}\text{MX}_4$, whose connected n layers of inorganic $[\text{MX}_6]$ octahedral monomolecular sheets are sandwiched by two organic barrier layers. The QWs are stacked together with atomically sharp interfaces and relatively weak van-der-Waals interactions. The excitons are quantum confined within the inorganic wells, but the quantum confinement effect decreases with increasing well width. Hence, the excitonic properties of the MQW can be systematically tailored from 2D to 3D by increasing the inorganic layer number from $n = 1$ to $n = \infty$ ¹⁸⁻²⁰. Recently, remarkable high efficiency perovskite LEDs constructed with the MQW assemblies have been demonstrated^{12,13}. However, the innate dynamics of the injected charge carriers in these MQWs assemblies driving the high efficiencies in these LEDs remain unclear.”

was changed to

*“In-between these 2D and 3D classes of perovskites, there is also a family of quasi-2D perovskite quantum wells (QWs) $L_2(\text{SMX}_3)_{n-1}\text{MX}_4$, whose connected n layers of inorganic $[\text{MX}_6]$ octahedral monomolecular sheets are sandwiched by two organic barrier layers. The QWs are stacked together with atomically sharp interfaces and relatively weak van-der-Waals interactions in the bulk crystal. The excitons are quantum confined within the inorganic wells, but the quantum confinement effect decreases with increasing well width. Hence, the excitonic properties of the quasi-2D QW can be systematically tailored from 2D to 3D by increasing the inorganic layer number from $n = 1$ to $n = \infty$ ²¹⁻²³. Tsai *et al.* reported that the*

excitonic properties of the high quality QW crystals ($n = 3$ and $n = 4$) is very close to that of the 3D perovskite ($n = \infty$) already²⁴. Recently, remarkable high efficiency perovskite LEDs constructed with the multi-QW (MQW) assemblies (with multi-thickness QWs in one crystal film) have been demonstrated^{14,15}. However, the innate dynamics of the injected charge carriers in these MQWs assemblies driving the high efficiencies in these LEDs remain unclear.”

New reference 24 **was added**:

24. Tsai, H. *et al.* High-efficiency two-dimensional Ruddlesden-Popper perovskite solar cells. *Nature* **536**, 312-316 (2016).

Reviewer 2

In the current manuscript the authors describe how to effectively overcome the low bimolecular recombination rate in perovskites by using 2-D, van-der-Waals assembled perovskite building blocks which results in favorable recombination when compared with 3-D bulk perovskites.

The manuscript is generally free of errors except for a lonely “)” in the abstract (line 21) and an “additional” on page 9, line 11. The references are adequate with a few exceptions as noted below.

The methods applied, the data gathered and the analysis performed are excellent. The findings are novel and provide a clear strategy into further improvement of light-emitting diodes incorporating this emitter materials. Therefore, the reviewer strongly recommends publication of this manuscript with very minor revisions.

We would like to thank the reviewer for the constructive comments. The typos pointed out by the referee have been corrected now.

1. In original manuscript, lines 20, 21: “*Under EL working conditions with typical charge densities $<10^{15} \text{ cm}^{-3}$,*” was changed to “**Under EL working conditions with typical charge densities $<10^{15} \text{ cm}^{-3}$,**”
2. In original manuscript, line 255: “*in additional to Auger recombination,*” was changed to “**in addition to Auger recombination,**”

1. *The only concern with the manuscript at the current stage is that it is very hard to extract the numbers that align with the title of the manuscript, e.g. how did the respective recombination rates change when move from 3D to 2D quantum wells; this information should be made accessible more easily, in a table for example as it is hard to extract from Figure 3d and the supporting information alone.*

We would like to thank the reviewer for the constructive comment. A table for these extracted parameters is now added and the manuscript revised accordingly.

1. In original Manuscript, lines 208-210: “*Figure 3a shows that the PL effective lifetime ($\tau_{PL}^{1/e}$) of the MQWs ($\sim 20 \text{ ns}$) is nearly two orders of magnitude shorter than that of the 3D counterpart ($>1 \mu\text{s}$),” **was changed to** “Figure 3a shows that the PL effective lifetime ($\tau_{PL}^{1/e}$) of the MQWs ($\sim 20 \text{ ns}$) is nearly two orders of magnitude shorter than that of the 3D counterpart ($>1 \mu\text{s}$) (Table 1),”*
2. In original Manuscript, lines 238, 239: “*The trap density associated with TQWs is determined to be ultra-low ($<3 \times 10^{13} \text{ cm}^{-3}$, Fig. S16).” **was changed to** “The trap density associated with TQWs is determined to be ultra-low ($<3 \times 10^{13} \text{ cm}^{-3}$, Fig. S16 & Table 1).”*

3. In original manuscript, lines 241-243: “the $\tau_{PL}^{1/e}$ of this excitonic recombination in TQWs is one to two orders of magnitude faster than that of the bimolecular recombination in 3D perovskites (Fig. 3c).” **was changed to** “the $\tau_{PL}^{1/e}$ of this excitonic recombination in TQWs is one to two orders of magnitude faster than that of the bimolecular recombination in 3D perovskites (Fig. 3c & Table 1).”
4. In original SI, lines 247, 248: “The above results also show that the biexciton recombination coefficient decreases with increasing the QW width.” **was changed to** “The above results also show that the biexciton recombination coefficient decreases with increasing the QW width (Table 1).”
5. In original SI, line 257: “and $k_3 = 3(\pm 2) \times 10^{-28} \text{ cm}^6 \text{ s}^{-1}$.” **was changed to** “and $k_3 = 3(\pm 2) \times 10^{-28} \text{ cm}^6 \text{ s}^{-1}$ (Table 1).”
6. In original SI, Page 7, lines 271, 272: “and $k_3 = 2(\pm 1) \times 10^{-26} \text{ cm}^6 \text{ s}^{-1}$.” **was changed to** “and $k_3 = 2(\pm 1) \times 10^{-26} \text{ cm}^6 \text{ s}^{-1}$ (Table 1).”
7. In original SI, lines 305-307: “the trap states density associated with the TQWs is estimated to be much lower than that of the 3D perovskite ($n_{TP}^{TQWs} < 3 \times 10^{13} \text{ cm}^{-3}$).” **was changed to** “the trap states density associated with the TQWs is estimated to be much lower than that of the 3D perovskite ($n_{TP}^{TQWs} < 3 \times 10^{13} \text{ cm}^{-3}$) (Table 1).”

At the end of the manuscript, Table 1 was **added**:

Table 1. The PL decay effective lifetime ($\tau_{PL}^{1/e}$) and quantum yield (η_{PL}) at injected carrier density of $\sim 10^{15} \text{ cm}^{-3}$, Trap states density (n_{Trap}), First order (k_1), Second order (k_2), and Third order (three-particle Auger) recombination constant (k_3), Biexciton Auger recombination coefficient (C_{bx}).

	$\tau_{PL}^{1/e}$ (ns)	η_{PL}	n_{Trap} (cm^{-3})	k_1 (s^{-1})	k_2 ($\text{cm}^3 \text{ s}^{-1}$)	k_3 ($\text{cm}^6 \text{ s}^{-1}$)	C_{bx} ($\text{cm}^2 \text{ s}^{-1}$)
3D	1300 ± 100	0.08 ± 0.02	3.5(± 0.8) $\times 10^{16}$	9(± 4) $\times 10^5$ (Trap-assisted)	7(± 3) $\times 10^{-10}$	3(± 2) $\times 10^{-28}$	-
MQWs	22 ± 2	0.60 ± 0.05	<3 $\times 10^{13}$ (a)	5(± 3) $\times 10^7$ (Excitonic)	2(± 0.6) $\times 10^{-9}$	2(± 1) $\times 10^{-26}$	-
2D ($n = 2$)	- (b)	- (b)	- (b)	- (b)	-	-	0.10 ± 0.02
2D ($n = 1$)	0.25 ± 0.03	0.02 ± 0.01	- (c)	4.0(± 0.4) $\times 10^9$ (Both)	-	-	0.44 ± 0.09

(a)The trap states density associated with the TQWs ($n \geq 5$) in the MQWs film.

(b)We could not achieve pure quasi-2D ($n = 2$) perovskite bulk crystal film.

(c)2D ($n = 1$) perovskite is not stable under high power laser excitation.

2. *With respect of the diffusion length of excitons in this materials I would recommend the authors to Brenner et al. (J. Phys. Chem. Lett., 2015, 6 (23), pp 4754–4757) and with respect to the electron-phonon coupling to Wright et al. (Nature Communications 7, Article number: 0 (2016) doi:10.1038/ncomms11755).*
3. *The authors missed a relevant reference related to early stage perovskite LEDs. The reviewer recommends to cite the following referenece in the introduction. [1] Y.-H. Kim et al, Adv. Mater., 27, 1248-1258 (2015)*

We would like to thank the reviewer for highlighting these recent relevant publications to us.

We have cited these references within our manuscript at the below locations:

1. In original manuscript, lines 35, 36: “long-range balanced charge carrier diffusion lengths⁴,” **was changed to** “long-range balanced charge carrier diffusion lengths^{4,5},”

New reference 5 **was added**:

5. Hodes, G., Kamat, P. V. Understanding the implication of carrier diffusion length in photovoltaic cells. *J. Phys. Chem. Lett.* **6**, 4090–4092 (2015).

2. In original manuscript, lines 43-45: “To date, respectable perovskite LED efficiencies (8.8%) with emission color tunable in a larger area than the National Television System Committee (NTSC) standard have been realized⁹⁻¹³.” **was changed to** “To date, respectable perovskite LED efficiencies (8.8%) with emission color tunable in a larger area than the National Television System Committee (NTSC) standard have been realized¹⁰⁻¹⁵.”

New reference 11 **was added**:

11. Kim, Y. H. et al. Multicolored organic/inorganic hybrid perovskite light-emitting diodes. *Adv. Mater.* **27**, 1248–1254 (2015).

3. In original SI, lines 174, 175: “~92.1 ± 23.5 meV for 3D CH₃NH₃PbI₃ perovskite⁴⁴.” **was changed to** “in a range of 40-92.1 meV for 3D perovskites (CH₃NH₃PbI₃, CH₃NH₃PbBr₃, CH₅N₂PbI₃, and CH₅N₂PbBr₃)^{50,51}.”

New reference 51 **was added**:

51. Wright, A. D. et al. Electron-phonon coupling in hybrid lead halide perovskites. *Nat. Commun.* **7**, 11755 (2016).

Reviewer 3

In this paper, Xing et al elucidated that van-der-Waals coupled perovskite multiple quantum-wells (MQWs) exhibit much faster and thus much more efficient radiative excitonic recombination, compared with conventional 3D perovskites or 2D perovskites. Such MQWs intrinsically overcome the bottleneck of the slow bimolecular recombination in lead-halide perovskites which limits the electroluminescent efficiency; very promising for high efficient electroluminescent applications. Using ultrafast time-resolved PL and transient techniques, they confirmed the very rapid localization (~0.5 ps) occurring from adjacent thin few layer ($n \leq 4$) MQW to the thick ($n \geq 5$) MQW with extremely high efficiency (>85%), avoiding conventional large defect trapping. The experimental results well support their conclusion and provide novel insight into the carrier dynamics in the perovskite MQWs. The manuscript is well organized and presented. This manuscript may be publishable if authors can address the following concerns and improve the manuscript:

1. The most conclusions are based on the comparison between MQWs and 2D (NMA)₂PbI₄ and 3D FAPbI₃ films. It is well known that compositions of perovskite significantly impact the carrier dynamics, in addition to structures. It is essentially important to justify the validity of the comparison;

We would like to thank the reviewer for the constructive comment. We agree with the reviewer that carrier dynamics in perovskites are also dependent on the chemical composition, not only the dimensional structure. Even with the same dimensional structure and same compositions, the decay parameters also vary from sample to sample due to different sample quality and size. However, as reported by Haiming *et al.* about the 3D MAPbBr₃, FAPbBr₃, and CsPbBr₃ perovskites, with the same dimensional structure and similar sample quality, different composition will result in slightly different decay parameters, but will not change the basic decay physics. Therefore, they reported that the first-order rate constants for carrier trapping are (4.0 ± 1.8) , (1.5 ± 0.8) , $(2.1 \pm 1.7) \times 10^7 \text{s}^{-1}$ and the second-order rate constants for electron-hole radiative recombination are (5.4 ± 1.8) , (3.5 ± 1.6) , $(5.7 \pm 2.3) \times 10^{-10} \text{cm}^3 \text{s}^{-1}$ for MAPbBr₃, FAPbBr₃, and CsPbBr₃, respectively (*Adv. Mater.* **2016**, DOI:10.1002/adma.201603027). Keeping the 3D structure of the lead-halide perovskite, the dominant radiative decay process will not change from free electron-hole recombination to excitonic recombination by varying only the composition of the perovskite (H. Zhu, *et al. Adv. Mater.* **2016**, DOI:10.1002/adma.201603027; L. M. Herz, *Annu. Rev. Phys. Chem.* **2016**, 67, 65-89.). The basic photophysics governing carrier decay are determined by the dimensional structure of the lead-halide perovskites, regardless of the composition.

In this manuscript, we focus on understanding the basic photophysical factors affecting the carrier recombination associated with the traditional 2D and 3D lead-halide perovskites for electroluminescence, and show how these limitations can be transcended with the self-

assembled MQWs. With 3D lead-halide perovskite, the radiative QY will always increase with increasing carrier density due to the gradual domination of the bimolecular recombination (second order) over the charge carrier trapping (first order), regardless of the sample composition. Therefore, it's extremely hard to achieve high luminescence QY at low carrier density (LED working condition) with the 3D lead-halide perovskites. With 2D ($n=1$) lead-halide perovskite bulk crystals, the significant nonradiative recombination (i.e., surface trapping, intrinsic defect trapping, extremely strong exciton-phonon and exciton-exciton interaction) is also persistent, regardless of the composition. Actually, high QY 2D ($n=1$) lead-halide perovskite bulk crystals have never been demonstrated.

Here, we show with fast and efficient charge carrier localization from thin QWs ($n \leq 4$) to thick QWs ($n \geq 5$) and excitonic type recombination in the thick QWs, the MQW assemblies possess near invariant high radiative QY at a large carrier density range, which covers the carrier densities typically used in LEDs. Therefore, based on the comparison between different kinds of physical decay processes associated with different dimensional structures, we come to the main conclusion: the slow bimolecular recombination limitation in 3D lead-halide perovskites for electroluminescence at relative low carrier density ($< 10^{15} \text{ cm}^{-3}$) can be effectively overcome by using these self-assembled perovskite MQWs.

The FAPbI_3 and $(\text{NMA})_2\text{PbI}_4$ are typical 3D and 2D ($n=1$) lead-halide perovskites. The optoelectronic properties of them presented here are consistent with previous reports (see the manuscript for details). We compared them with $(\text{NMA})_2\text{FAPb}_2\text{I}_7$ (MQWs) to illustrate the difference between different decay processes. However, the conclusions stated above are rather general, not limited to these systems.

In view of this concern raised by the reviewer, we have added the following justification to the manuscript:

1. In original manuscript, lines 102-105:

*“In this study, the perovskite MQWs film was deposited with a precursor solution of 1-naphthylmethylamine iodide (NMAI, $\text{C}_{10}\text{H}_7\text{CH}_2\text{NH}_3\text{I}$), formamidinium iodide (FAI, CH_5IN_2) and PbI_2 with a molar ratio of 2:1:2 dissolved in *N,N*-dimethylformamide (DMF)^{18,19}.”*

was changed to

“In this study, the perovskite MQWs film was deposited with a precursor solution of 1-naphthylmethylamine iodide (NMAI, $\text{C}_{10}\text{H}_7\text{CH}_2\text{NH}_3\text{I}$), formamidinium iodide (FAI, CH_5IN_2)

and PbI_2 with a molar ratio of 2:1:2 dissolved in N,N-dimethylformamide (DMF)^{21,22}. With the same dimensional structure and similar sample quality of the lead-halide perovskites, different composition will result in slightly different decay parameters, but will not change the basic decay physics^{30,31}. Therefore, the photo physics illustrated below is broadly applicable.”

New references 30 and 31 **were added**:

30. Zhu, H., Trinh, M. T., Wang, J., Fu, Y., Joshi, P. P., Miyata, K., Jin, S., Zhu, X. Y. Organic cations might not be essential to the remarkable properties of band edge carriers in lead halide perovskites. *Adv. Mater.* DOI:10.1002/adma.201603072 (2016).
31. Herz, L. M. Charge-carrier dynamics in organic-inorganic metal halide perovskites. *Annu. Rev. Phys. Chem.* **67**, 65-89. (2016).

2. In original manuscript, lines 258-260:

“Nevertheless, the MQWs show a nearly invariant high QY excitonic recombination at carrier density below $1.5 \times 10^{16} \text{ cm}^{-3}$. In contrast, the luminescence QY of the bimolecular recombination in 3D perovskite significantly decreases with decreasing the injected carrier density (Fig. 3d).”

was changed to

“Nevertheless, the MQWs show a nearly invariant high QY excitonic recombination at carrier density below $1.5 \times 10^{16} \text{ cm}^{-3}$. In contrast, the luminescence QY of the bimolecular recombination in 3D perovskite significantly decreases with decreasing the injected carrier density (Fig. 3d). The absolute luminescence QY value may slightly vary with the chemical composition of the perovskite. However, the dependence of the luminescence QY on carrier density should persist since it is governed by the basic carrier decay physics, which is determined by the structural dimension of the perovskite, regardless of the composition.”

2. It is not evident that the perovskite MQWs remain high charge carrier mobility of halide perovskite. So it is not a suitable conclusion, particularly in abstract;

We thank the reviewer for pointing out our oversight. Here, we missed the references for this claim.

In original manuscript, lines 79-82: we stated “Recently, remarkable high efficiency perovskite LEDs constructed with the MQW assemblies have been demonstrated^{12,13}.”

However, the innate dynamics of the injected charge carriers in these MQWs assemblies driving the high efficiencies in these LEDs remain unclear.”

In both original refs. 12 and 13 (new version refs. 14, 15), the charge carrier mobility of the perovskite MQW assemblies is shown to decrease from the value of 3D perovskite to the value of 2D ($n=1$) perovskite with increasing the long chain organic cation percentage from 0 to 1 (**Figure R1**). Therefore, the carrier mobility of the MQW assemblies should be limited by the charge carrier transporting capability of the dominant thin QWs inside. Since high charge carrier mobility of the 2D ($n=1$) perovskite has been reported already (original version Refs 22, 23), that’s why we claimed the perovskite MQWs remain high charge carrier mobility and stated “*These perovskite thin QWs inherit the favorable properties of 2D perovskites: simple low-temperature solution processability, high charge carrier mobility and uniform morphology*^{22,23}.” in original manuscript, lines 95-97. We **have now provided proper references** to support this claim:

“These perovskite thin QWs inherit the favorable properties of 2D perovskites: simple low-temperature solution processability, high charge carrier mobility and uniform morphology^{14,15,26,27}.”

For clarification and accuracy, we have revised the abstract accordingly:

Original manuscript, lines 27, 28: “*These MQWs retain the simple solution processability and high charge carrier mobility of halide perovskites.*” **was changed to** “These MQWs retain the simple solution processability and high charge carrier mobility of **2D lead-halide perovskites.**”

Figure R1: Current density versus voltage of the perovskite MQWs LEDs. **(Left)** ITO/Buf-HIL (50 nm)/perovskite film/TPBI (50 nm)/LiF (1 nm)/Al (100 nm) structure. Adopted from [Adv. Mater. **28**, 7515 (2016), Fig. 5a]. **(Right)** ITO/TiO₂/(PEA₂(MA)_{n-1}Pb_nI_{3n+1})/F8-MoO₃/Au structure. Adopted from [Nat. Nanotechnol. **11**, 872 (2016), Fig. S21].

3. When discussing 2D perovskites, the statement originates from the literatures published long time ago. Due to recent significant progress (for example, Dou et al in *Science* and Tsai et al in *Nature*), the statement may not really suitable.

We agree with the reviewer that significant progress has been made recently in preparation of high quality 2D ($n = 1$) and quasi-2D ($n \geq 2$) lead-halide perovskite.

Dou et al. showed high quality single unit cell or a few unit cells thick 2D ($n = 1$) lead-halide perovskite sheets could be prepared with a ternary co-solvent method [*Science* **349**, 1518 (2015)]. Such atomically thin 2D sheets are promising for nano-scale optoelectronic devices. However, they also clearly stated that “The PLQE for the 2D sheet is calculated to be ~26%, which is much higher than the QE of the bulk crystal (<1%)” [Page 1520, middle column, lines 20-21]. The PLQY of 2D ($n = 1$) bulk crystal is much lower than that of thin 2D ($n = 1$) sheet. The high quality 2D ($n = 1$) bulk crystal remains to be discovered, which is the essential part for constructing a traditional macroscopic high efficiency 2D perovskite LEDs.

Tsai et al. prepared high quality quasi-2D $(\text{BA})_2(\text{MA})_2\text{Pb}_3\text{I}_{10}$ ($n = 3$) and $(\text{BA})_2(\text{MA})_3\text{Pb}_4\text{I}_{13}$ ($n = 4$) perovskite bulk crystal with a hot-casting technique [*Nature* **536**, 312 (2016)]. They demonstrated that the solar cell devices constructed with these quasi-2D perovskite show much better environmental stability and photostability than the 3D counterpart (MAPbI_3) due to the passivation by the long BA^+ cation. According to their claims, the charge carrier dynamics in these quasi-2D perovskites should also be dominated by free charge carriers (Page 313, column 1, lines 7-10; column 2, lines 1-5, **see below adaptations**) and experience similar trap states density ($\sim 10^{16} \text{cm}^{-3}$, Page 315, column 1, lines 35,36) as that in the 3D perovskites. Therefore, these quasi-2D perovskite crystals would face the same “bimolecular recombination limitation” as discussed in our manuscript.

[*Nature* **536**, 312 (2016), Page 313, column 1, lines 7-10; column 2, lines 1-5]: “Furthermore, we predict that for $(\text{BA})_2(\text{MA})_2\text{Pb}_3\text{I}_{10}$ ($n = 3$) and $(\text{BA})_2(\text{MA})_3\text{Pb}_4\text{I}_{13}$ ($n = 4$) compounds, the exciton binding energy is closer to that of MAPbI_3 ($n \rightarrow \infty$), for which the excitons are expected to be almost ionized at room temperature and charge-carrier transport is expected to be dominated by free carriers (Supplementary Discussion). These theoretical predictions are in good agreement with the experimentally measured optical absorption spectra, which do not exhibit excitonic signatures (Fig. 1e).”

Therefore, the discussions related to 2D perovskite in the manuscript have been revised:

1. In original manuscript, lines 59-63: “*Although 2D perovskites associated with fast excitonic emission and high charge carrier mobility are promising, daunting challenges pertaining to significant nonradiative recombination need to be effectively tackled (i.e., surface trapping, intrinsic defect trapping, extremely strong exciton- phonon coupling and exciton-exciton interaction; see SI).*” **was changed to** “*Although 2D perovskite bulk crystals²⁰ associated with fast excitonic emission and high charge carrier mobility are promising, daunting challenges pertaining to significant nonradiative recombination need to be effectively tackled (i.e., surface trapping, intrinsic defect trapping, extremely strong exciton- phonon coupling and exciton-exciton interaction; see SI).*”

New reference 20 **was added**:

20. Dou, L. *et al.* Atomically thin two-dimensional organic-inorganic hybrid perovskites. *Science* **349**, 1518-1521 (2015).

2. In original manuscript, lines 67-69: “*The injected excitons are therefore tightly confined in the 2D inorganic layers and feature large exciton binding energies and undergo significant non-radiative recombination¹⁶⁻²⁰.*” **was changed to** “*The injected excitons are therefore tightly confined in the 2D ($n = 1$) inorganic layers and feature large exciton binding energies and undergo significant non-radiative recombination in the 2D bulk crystals¹⁸⁻²³.*”

3. In original manuscript, lines 72-82:

“In-between these 2D and 3D classes of perovskites, there is also a family of perovskite multi-quantum-well (MQW) crystals $L_2(SMX_3)_{n-1}MX_4$, whose connected n layers of inorganic $[MX_6]$ octahedral monomolecular sheets are sandwiched by two organic barrier layers. The QWs are stacked together with atomically sharp interfaces and relatively weak van-der-Waals interactions. The excitons are quantum confined within the inorganic wells, but the quantum confinement effect decreases with increasing well width. Hence, the excitonic properties of the MQW can be systematically tailored from 2D to 3D by increasing the inorganic layer number from $n = 1$ to $n = \infty$ ¹⁸⁻²⁰. Recently, remarkable high efficiency perovskite LEDs constructed with the MQW assemblies have been demonstrated^{12,13}. However, the innate

dynamics of the injected charge carriers in these MQWs assemblies driving the high efficiencies in these LEDs remain unclear.”

was changed to

“In-between these 2D and 3D classes of perovskites, there is also a family of quasi-2D perovskite quantum wells (QWs) $L_2(\text{SMX}_3)_{n-1}\text{MX}_4$, whose connected n layers of inorganic $[\text{MX}_6]$ octahedral monomolecular sheets are sandwiched by two organic barrier layers. The QWs are stacked together with atomically sharp interfaces and relatively weak van-der-Waals interactions in the bulk crystal. The excitons are quantum confined within the inorganic wells, but the quantum confinement effect decreases with increasing well width. Hence, the excitonic properties of the quasi-2D QW can be systematically tailored from 2D to 3D by increasing the inorganic layer number from $n = 1$ to $n = \infty$ ²¹⁻²³. Tsai *et al.* reported that the excitonic properties of the high quality QW crystals ($n = 3$ and $n = 4$) is very close to that of the 3D perovskite ($n = \infty$) already²⁴. Recently, remarkable high efficiency perovskite LEDs constructed with the multi-QW (MQW) assemblies (with multi-thickness QWs in one crystal film) have been demonstrated^{14,15}. However, the innate dynamics of the injected charge carriers in these MQWs assemblies driving the high efficiencies in these LEDs remain unclear.”

New reference 24 **was added**:

24. Tsai, H. *et al.* High-efficiency two-dimensional Ruddlesden-Popper perovskite solar cells. *Nature* **536**, 312-316 (2016).

4. In original SI, lines 99,100: “*The direct-bandgap lead-halide single layer 2D perovskites are monomolecular layers of $[\text{PbI}_6]$ octahedral sandwiched between long organic barrier layers.*” **was changed to** “The direct-bandgap lead-halide single layer 2D ($n = 1$) perovskites are monomolecular layers of $[\text{PbI}_6]$ octahedral sandwiched between long organic barrier layers.”
5. In original SI, lines 106,107: “*However, due to the presence of defects and some other unclear reasons, the QY in lead-halide 2D perovskites has typically been very poor*^{16-20,22,23}.” **was changed to** “However, due to the presence of defects and some other unclear reasons, the QY in lead-halide 2D ($n = 1$) perovskite bulk crystals has typically been very poor^{18-23,26,27}.”

4. Line 176 “Figures 1c & S12 clearly show that over 95% of PL from the MQWs is dominated by the recombination in TQWs while the excitons are primarily injected into bilayers with density $<10^{16} \text{ cm}^{-3}$.”. The 95% is not clearly shown and inconsistent to FigS12d.

We would like to thank the reviewer for the constructive comment. To present the results more clearly, we have revised Fig. S12c and its caption.

Original Fig. S12:

Figure S12 | Carrier density dependent exciton localization from thin QWs to TQWs. (a) Pump fluence dependent PL of the MQWs film. (b) The dependence of PL peak intensity on photon-injected carrier density at the exciton resonance of the QWs with different inorganic layers. (c) The injected carrier density dependent PL peak intensity ratios of bilayer and trilayer QW to TQW. (d) The extracted injected carrier density dependence of charge transfer efficiency from thin QW to TQW. The inset shows detailed fitting of the dependence of integrated PL intensity on the injected carrier density.

was changed to

Figure S12 | Carrier density dependent exciton localization from thin QWs to TQWs. (a) Pump fluence dependent PL of the MQWs film. (b) The dependence of PL peak intensity on photon-injected carrier density at the exciton resonance of the QWs with different inorganic layers. (c) The injected carrier density dependent PL peak intensity ratios of bilayer and trilayer QW to TQW. The inset shows the PL spectrum could be well fitted with 5 Gaussian peaks. The two dominant broad peaks are attributed to emission from the TQWs ($n \geq 5$), which occupy 96.3% of the total emission intensity at carrier density of $\sim 1.0 \times 10^{16} \text{ cm}^{-3}$. (d) The extracted injected carrier density dependence of charge transfer efficiency from thin QW to TQW. The inset shows detailed fitting of the dependence of integrated PL intensity on the injected carrier density.

Here we would like to clarify that Fig. S12d shows the injected carrier density dependent charge transfer efficiency from thin QWs to TQWs. If the light emission QY of the thin QWs is the same as that of the TQWs, then the light emission intensity ratio of the TQWs to the whole emission spectrum should be the same value as the charge transfer efficiency. However, due to the light emission QY of the TQWs being much higher than that of the thin QWs, therefore the charge transfer efficiency ($>85\%$) from thin QWs to TQWs is lower than the light emission intensity percentage ($>95\%$) dominated by the TQWs at carrier densities below $1.0 \times 10^{16} \text{ cm}^{-3}$. There is no inconsistency between these results.

5. Line 293 “Room temperature relative PLQY at different excitation intensity of the perovskite ...” Here PLQY is incorrect because you actually do not take into account the different absorption and scattering.

We thank the reviewer for this comment. We agree with the reviewer that different absorption and scattering should be considered for absolute PLQY measurements. However, for measuring the relative PLQY of sample at different excitation intensity, all the experimental conditions are fixed except for the excitation power, which is controlled by adjusting neutral density filters. In the pump fluence range of linear light-matter interaction, with fixed absorption coefficient, refractive index and geometry of the sample, the light absorbed by the sample should linearly increase with increasing the excitation light intensity. With fixed PL collection efficiency, the relative PLQY at different excitation intensity should be proportional to the collected PL intensity normalized by the excitation light intensity. At extremely high pump fluence, nonlinear absorption and sample damage will introduce uncertainty to this method. With small power (<3mW, 2.5 MHz) and long duration (~200 ps) pulses, the nonlinear absorption observed typically is quite small.

For clarity, we have revised this part accordingly. In original manuscript, lines 293-301:

“Room temperature relative PLQY at different excitation intensity of the perovskite thin films was measured with a time-resolved confocal microscope (MicroTime 200, PicoQuant). The samples were excited with 405 nm (200 ps, 2.5 MHz) laser pulses generated from the LDH-P-C-405B laser diode (PicoQuant). The luminescence was then passed through a dichroic mirror (T425lpxr, Chroma) and an 715 nm long pass filter before being detected by a single-photon avalanche diode (SPAD, SPCM-AQR-15, Perkin-Elmer). A three-step technique was used to obtain the absolute PLQY values of perovskite films by combination of a 445 nm continuous wave (CW) laser, optical fiber, spectrometer and integrating sphere³¹.”

was changed to

“Room temperature relative PLQY at different excitation intensity of the perovskite thin films was measured with a time-resolved confocal microscope (MicroTime 200, PicoQuant). The samples were excited with 405 nm (200 ps, 2.5 MHz, <3 mW) laser pulses generated from the LDH-P-C-405B laser diode (PicoQuant). The luminescence was then passed through a dichroic mirror (T425lpxr, Chroma) and an 715 nm long pass filter before being detected by a single-photon avalanche diode (SPAD, SPCM-AQR-15, Perkin-Elmer). The PL at different excitation intensity was recorded under same exact experimental conditions

except for the excitation intensity, which is controlled by adjusting the neutral density filters. Under the assumption of negligible nonlinear absorption and negligible sample damage, the relative PLQY at different excitation intensity is obtained by normalizing the collected PL intensity with the corresponding excitation light intensity. The relative PLQY at high carrier density ($>10^{17} \text{ cm}^{-3}$) may be slightly underestimated due to the small saturable absorption of the samples at 405 nm. A three-step technique was used to obtain the absolute PLQY values of perovskite films by combination of a 445 nm continuous wave (CW) laser, optical fiber, spectrometer and integrating sphere³⁷.”

6. *Fig S19a may be not valid, given very low intensity.*

We would like to thank the reviewer for this constructive comment. Under the same experimental conditions, the 2D ($n=1$) perovskite PL intensity is much lower (**darker**) and PL lifetime is much shorter (more **blue**) than that of the self-assemble MQWs or 3D perovskite. These comparisons can be clearly shown with Fig. S19 under the same scale bars. However, the low PL intensity of the 2D perovskite may induce large uncertainty to the estimated PL intensity standard deviations. The light emission inhomogeneity of the 2D perovskite may be overestimated. As pointed by the reviewer, we have revised the manuscript accordingly.

In original SI, lines 328-330: “*The spatial homogeneity shows an integrated PL intensity standard deviation of $\pm 10\%$, $\pm 20\%$ and $\pm 39\%$ for the MQWs, 2D and 3D perovskite films, respectively.*” **was changed to** “The spatial homogeneity shows an integrated PL intensity standard deviation of $\pm 10\%$, $\pm 20\%$ and $\pm 39\%$ for the MQWs, 2D and 3D perovskite films, respectively. **The light emission inhomogeneity of the 2D perovskite film may be overestimated due to the low PL intensity collected.**”

REVIEWERS' COMMENTS:

Reviewer #1 (Remarks to the Author):

I confirmed the previous review comments by all referees, the replies by the authors, and the revised manuscript. The resubmitted manuscript has been well improved and its goal is clear. The authors conduct suitable and careful photoluminescence experiments and analyze the results. The conclusions drawn from this analysis appear to be robust. However, the manuscript title is "for electroluminescence" and high efficiency perovskite light-emitting diodes were reported in Refs 14 and 15. In my opinion, although this resubmitted manuscript is timely in our community, I do not think that this work merits publication in Nature Communications, from a viewpoint of impact and novelty of the revised manuscript.

Reviewer #2 (Remarks to the Author):

The revised version of the manuscript titled "Transcending the Slow Bimolecular Recombination in Lead-Halide Perovskites for Electroluminescence" has adequately addressed all of the previously raised questions. The overall quality of the manuscript in terms of content and scientific presentation has improved to a point that allows for publication in nature communications.

Reviewer #4 (Remarks to the Author):

Authors have carefully considered and addressed the reviewers' comments; accordingly, they revise and improve the manuscript. I recommend publishing this work.

Reviewer's comments to the author

Reviewer 1

I confirmed the previous review comments by all referees, the replies by the authors, and the revised manuscript. The resubmitted manuscript has been well improved and its goal is clear. The authors conduct suitable and careful photoluminescence experiments and analyze the results. The conclusions drawn from this analysis appear to be robust. However, the manuscript title is "for electroluminescence" and high efficiency perovskite light-emitting diodes were reported in Refs 14 and 15. In my opinion, although this resubmitted manuscript is timely in our community, I do not think that this work merits publication in Nature Communications, from a viewpoint of impact and novelty of the revised manuscript.

We would like to thank the reviewer for his further comments. Indeed, metal-halide perovskites have recently attracted a lot attention for development in light emission applications. Perovskites of various dimensional (including 2D bulk crystal, 3D bulk crystal, nanocrystals, nanocrystals embedded in conduction polymers, multi-quantum wells (MQWs), etc.) have been tested for both lasing and electroluminescence. Respectable light emission efficiencies and technological advancements have been achieved. However, the basic photophysics associated with these perovskites of various dimensions remains unclear. These are critical knowledge which can spur or limit the light emission efficiency. Hence, a clear understanding is absolutely essential for further optimization of the device efficiency. In our manuscript, we had clearly pointed out what are the fundamental limitations for the traditional 2D, 3D and nanocrystal perovskites for electroluminescence, and how these limitations could be overcome by using the MQWs. To go into the details, we would like to repeat some of our previous rebuttal below:

Most recently, remarkably high efficiency perovskite light emitting diodes (LEDs) constructed with the MQW assemblies have been demonstrated (refs. 14, 15). However, the innate dynamics of the injected charge carriers in these MQWs assemblies driving the high efficiencies in these LEDs remain unclear. We would like to clarify that in this manuscript, we advance the fundamental photophysical knowledge into overcoming the intrinsic limitations of 2D and 3D perovskites and towards achieving high efficiency perovskite LEDs. In this work, we clearly rationalize how these limitations in 2D and 3D perovskites are overcome with the Van-der-Waals coupled 2D-3D perovskite mixtures (self-assembled MQWs) through detailed femtosecond transient optical spectroscopy. The MQWs bypass the significant non-radiative recombination faced by the 2D perovskite with quantum coupling-induced ultrafast charge localization from thin QWs (inorganic layer number $n \leq 4$) to thick (T) QWs ($n \geq 5$). The relatively weak van-der-Waals coupling between adjacent QWs rendered the TQWs electronically isolated and passivated. The self-assembled TQWs inherit the high crystallinity

and low trap density of perovskite nanocrystals. The fast and efficient excitonic recombination in the TQWs allow the MQW assemblies to effectively overcome the slow bimolecular recombination bottleneck for 3D perovskite LEDs. These complementary properties in van-der-Waals coupled MQWs which are typical of epitaxial vacuum-grown heterojunctions, render us new functionalities that are not available with traditional halide perovskite single phase constituents. More specifically the key points from our study are:

- 1. First report on the clear photophysical picture of the fundamental limitations in the 3D and 2D lead-halide perovskites for LEDs.*
- 2. First report presenting concrete evidences (experimental and modelling) on how the limitations in 3D and 2D perovskites are overcome by using the perovskite MQWs. Although high efficiency perovskite LEDs constructed with the MQW assemblies have been demonstrated, the innate dynamics of the injected charge species (free-carrier or excitonic recombination) in these MQWs assemblies driving the high efficiencies in these LEDs remain unknown.*
- 3. First report on how to achieve the emission color tunability (dual emission) with weakly bound 2D perovskite stackers as well as detailed insights into QW-width dependent charge carrier dynamics.*

In view of the reviewer's concerns and for better clarity and easy accessibility of all the discussion and data for comparison, we have also moved the following two sections from the supplementary information into the main text:

In original SI, section 2:

“2. The slow bimolecular recombination limitation for using 3D lead-based perovskite in electroluminescence (EL)”

was moved into the main text as part of the discussion.

We hope that our justifications and efforts could fully address the referee's concerns.

Reviewer 2

The revised version of the manuscript titled "Transcending the Slow Bimolecular Recombination in Lead-Halide Perovskites for Electroluminescence" has adequately addressed all of the previously raised questions. The overall quality of the manuscript in terms of content and scientific presentation has improved to a point that allows for publication in nature communications.

We would like to thank the reviewer for the positive comments and support.

Reviewer 4

Authors have carefully considered and addressed the reviewers' comments; accordingly, they revise and improve the manuscript. I recommend publishing this work.

We would like to thank the reviewer for the positive comments and support.